# Evaluation of Poly-3-Hydroxybutyrate (P3HB) Scaffolds Used for Epidermal Cells Growth as Potential Biomatrix

**DOI:** 10.3390/polym14194021

**Published:** 2022-09-26

**Authors:** Sandra García-Cerna, Uriel Sánchez-Pacheco, Angélica Meneses-Acosta, José Rojas-García, Bernardo Campillo-Illanes, Daniel Segura-González, Carlos Peña-Malacara

**Affiliations:** 1Laboratorio 7 de la Facultad de Farmacia, Universidad Autonoma del Estado de Morelos, Avenida Universidad No. 1001, Chamilpa, Cuernavaca C.P. 62209, Morelos, Mexico; 2CIATEQ A. C. Plasticos y Materiales Avanzados. Av. Del Retablo 150, Queretaro C.P. 76150, Queretaro, Mexico; 3Instituto de Ciencias Fisicas, Universidad Nacional Autonoma de Mexico, Avenida Universidad S/N, Chamilpa, Cuernavaca C.P. 62210, Morelos, Mexico; 4Departamento de Ingenieria Celular y Biocatalisis, Instituto de Biotecnologia, Universidad Nacional Autonoma de Mexico, Apdo. Post. 510-3, Cuernavaca C.P. 62250, Morelos, Mexico

**Keywords:** P3HB scaffolds, tissue engineering, keratinocyte, 3D printing

## Abstract

Advances in tissue engineering have made possible the construction of organs and tissues with the use of biomaterials and cells. Three important elements are considered: a specific cell culture, an adequate environment, and a scaffold. The present study aimed to develop P3HB scaffolds by 3D printing and evaluate their biocompatibility with HaCaT epidermal cells, as a potential model that allows the formation of functional tissue. By using a method of extraction and purification with ethanol and acetone, a biopolymer having suitable properties for use as a tissue support was obtained. This polymer exhibited a higher molecular weight (1500 kDa) and lower contact angle (less than 90°) compared to the material obtained using the conventional method. The biocompatibility analysis reveals that the scaffold obtained using the ethanol–acetone method and produced by 3D printing without pores was not cytotoxic, did not self-degrade, and allowed high homogenous cell proliferation of HaCaT cells. In summary, it is possible to conclude that the P3HB scaffold obtained by 3D printing and a simplified extraction method is a suitable support for the homogeneous development of HaCaT keratinocyte cell lineage, which would allow the evaluation of this material to be used as a biomatrix for tissue engineering.

## 1. Introduction

Tissue engineering is the process of generating new tissues to replace diseased or injured ones. This technology uses biodegradable three-dimensional scaffolds to mimic the structure and functions of the natural extracellular matrix, in vitro tissue harvesting, and expansion methods [1,2,3]. The scaffolds are structures designed to be implanted within a living system to promote the regeneration of damaged tissue using the same type of cells. These structures must be designed to support cell adhesion, proliferation, and differentiation [3,4], as well as to be biocompatible and biodegradable to avoid any adverse effects.

Biomaterials, such as natural polymers (derived from renewable sources, such as plants, animals, and microorganisms), have frequently been used for tissue engineering applications, because they can have similar properties to the extracellular matrix, such as pseudoplastic behaviour, gelation capacity, water retention capacity, and biodegradability, among others [5,6,7].

Poly-3-hydroxybutyrate (P3HB) is a natural polymer that belongs to the family of poly-3-hydroxyalkanoates (PHAs), composed of 3-hydroxy fatty acids that can be produced by bacteria, such as *Zoogloea ramigera, Ralstonia eutropha, Methylobacterium, Escherichia coli, Pseudomonas acidophilia, Pseudomonas oleovorans, Pseudomonas aeruginosa, Pseudomonas putida,* and *Azotobacter vinelandii* [8,9]. This polymer is sustainable, versatile, biocompatible, and bioresorbable, and is, therefore, a suitable material for use in the pharmaceutical and biomedical fields. The molecular weight of P3HB is a chemical characteristic that, together with the monomeric composition and purity, determines the physicochemical and thermomechanical properties, such as the degree of crystallinity and mechanical resistance, and, consequently, their applications [10,11,12].

It is important to point out that P3HB with a molecular weight of less than 1000 kDa, produces a rigid and brittle material, due to its high degree of crystallinity (between 55 and 80%), and has faster thermal degradation than P3HB of high molecular weight (>1 × 10^3^ kDa), which has better elastic properties [11,13]. For this reason, P3HB can be used in the biomedical area to produce suture material, swabs, bandages, drug encapsulation systems, and supports for the culture of cells of cardiovascular, nervous, and dermal tissue [14,15,16].

Previous studies by García et al. [17] have shown that it was possible to obtain a polymer (P3HB) with a high molecular mass (5693 kDa), a high purity 95%) and high yields (85%), using an extraction method base on ethanol/acetone. In contrast, with the conventional method, based on the extraction with chloroform, the P3HB isolated had a molecular mass of 547 kDa.

It is important to point out that the final characteristics and properties of the scaffolds used for tissue engineering are determined by the fabrication technique used, such as the control of topography and surface roughness, porosity, shape, and pore size [18]. Currently, there are several techniques for the design of scaffolds with multiple shapes, textures, and biomaterials. These techniques range from the conventional, such as solvent dissolution-particle leaching, cold drying, thermally induced phase separation, solution-gel, and gas foaming, which have the disadvantage of using organic solvents for scaffold formation that could be toxic for cell culture and require more purification steps [19,20], to novel techniques, such as stereolithography, melted deposition modelling, selective laser, and electrospinning, which have the disadvantage of not conferring a homogeneous surface to the scaffold, causing the heterogeneous proliferation of cells [12,20].

3D printing, since the first printer was invented in 1987, has gained in relevance in this field due to the multiple advantages that this technology offers over the previously mentioned techniques. This method allows the generation of versatile scaffolds with complex shapes, which promote a homogeneous cell distribution, and product customization according to the biological system to be developed or the biomedical application. In addition, this technology can be used to design one or two-component scaffolds with homogeneous pore sizes and high interconnection for cell migration and transfer of the culture medium [21,22,23].

Therefore, the objective of this study was to obtain a biopolymer, using non-halogenated compounds, such as ethanol and acetone, and to design P3HB scaffolds by the 3D printing technique, testing their biocompatibility using HaCaT epidermal cells as a model to develop a functional biomatrix with potential use in tissue engineering.

## 2. Materials and Methods

### 2.1. Microorganism and Culture Conditions

The *Azotobacter vinelandii* strain OPNA (NRRL B-50898), a P3HB overproducing mutant [24], was used to produce this polymer. This strain was propagated and maintained by monthly subculture on peptone-yeast extract (PY) agar with spectinomycin (12.5 µg/L) and kanamycin (1.5 μg/L).

### 2.2. Culture Medium and Culture Conditions for the P3HB Production

The culture medium used for the batch cultures contained sucrose and yeast extract (BD Bac-to) as the nitrogen and carbon sources in the concentrations previously described [17]. Fed-batch cultures were carried out in a 500 L bioreactor (Made in México), with a final 350 L working volume. The aeration rate was 1 vvm (2 L/min) and two agitation rates were used: 500 rpm during the batch stage and 700 rpm when the feeding stage started (22 h). The pH was kept constant at 7.2 and controlled by the addition of NaOH (2 M) or HCl (1 M) solution through an on–off system using a peristaltic pump. The temperature was maintained at 29 °C and DOT was measured with an Ingold polarographic probe (Mettler-Toledo, Columbus, OH, USA). Nutrient pulses of 100 mL of a solution of sucrose (30 g/L) and yeast extract (15 g/L) were added to these cultures at different times of cultivation (22 and 39 h), when the sucrose concentration dropped to 10 g/L, to avoid sucrose limitation [17].

### 2.3. Analytical Methods

Microbial growth to produce P3HB was evaluated through protein measurements by the Lowry method using bovine serum albumin as a standard [25]. The total biomass dry weight was measured as described previously [24] and the sucrose concentration was evaluated by DNS reaction [26].

### 2.4. Separation Methods Used

#### 2.4.1. Rupture and Extraction with Ethanol-Acetone

This method was previously reported [17] and is based on the breaking of the cells using ethanol at its boiling point (77 ± 1 °C) and acetone, and their subsequent drying [17].

#### 2.4.2. Rupture and Extraction with Hypochlorite-Chloroform

For P3HB extraction using the hypochlorite-chloroform method, the system of two phases reported previously [27] was employed.

#### 2.4.3. P3HB Quantification

Determination of P3HB was made by high-performance liquid chromatography (HPLC, Waters Alliance 2695, Milford, MA, USA) after its conversion into crotonic acid by acidic hydrolysis with H_2_SO_4_. Samples of 3–5 mg of the biomass or P3HB after extraction treatments, were hydrolysed with 1 mL of H_2_SO_4_ at 90 °C for 1 h. Hydrolysed samples were diluted (1/40 with Milli-Q water) and injected (20 μL) into HPLC equipment (Separation module Alliance 2695, Waters, MA, USA) with an Aminex HPX-87H column (Biorad, Hercules, CA, USA). The eluent was H_2_SO_4_ (5 mmol/L) at 50 °C with a flow rate of 0.65 mL/min. P3HB converted into crotonic acid was detected with a Diode array detector at 220 nm.

#### 2.4.4. Yield and Purity Determination of P3HB Extracted from Cultivations

Equations (1) and (2) were used to estimate the recovery and purity yields.

(1)Recovery Yield (%) = (g P3HB solid extracted/g P3HB in biomass) × 100

(2)Purity (%) = (g P3HB/g P3HB solid extracted) × 100

#### 2.4.5. Molecular Weight Determinations

The P3HB molecular weight distribution was determined by gel permeation chromatography (GPC), using the SHODEX GPCK-806M column (Showa Denko, Tokyo, Japan) [28]. HPLC equipment (Waters Alliance 2695, Milford, MA, USA), with a refractive index detector (Waters, 2414, Milford, MA, USA), was used for this analysis. The operating conditions were as follows: a temperature of 30 °C, a flow rate of 0.7 mL/min chloroform (filtrate) as the mobile phase, 50 µL injection volume, and an 18 min run time. Polystyrene standards were used for the calibration curve with molecular weights between 2.9 and 5.97 × 10^3^ kDa. 15 mg of P3HB obtained using the different extraction methods were weighed and dissolved in 1 mL of chloroform for 2 or 3 h at 55 °C and 700 rpm in a thermoblock (Thermomixer R Eppendorf, Hamburg, Germany). Each sample was filtered through a 0.45 µm nylon membrane and placed in an HPLC vial. Empower software was used for sample processing and quantification.

#### 2.4.6. FTIR Characterization

Infrared spectra were obtained with a Nicolet iS10 FTIR spectrometer from Thermo Scientific (Waltman, MA, USA). P3HB samples obtained by the conventional method, as well as that obtained by the ethanol–acetone method were analysed, for each spectrum scans were performed between 4000 to 400 cm^−1^, with a resolution of 4 cm^−1^ [12].

### 2.5. Design of P3HB Scaffolds by 3D Printing

Circular scaffold models with a diameter of 1 cm and a thickness of 100 µm were produced using design software (AutoCAD), and P3HB was isolated using the ethanol–acetone method. A 3D printer was fed with the P3HB filament at 40 mm/s. The nozzle was heated to 185 °C and the print bed was heated to 60 °C to generate the scaffolds. The printed P3HB scaffolds were sterilized using ethylene oxide and washed with sterile PBS-Antibiotic-Antimycotic 100× solutions (5, 4, 3, 2, and 1% *v*/*v*) for 5 min in each solution. After washing, the P3HB scaffolds were stored in Falcon tubes under 50 mL of sterile PBS.

### 2.6. Morphological Analysis and Contact Angle Measurements

Morphological analysis of the scaffolds generated by 3D printing was performed by scanning electron microscopy (SEM). Images were taken at 15 kV at 300×, 1000× and 4000× magnifications. For this purpose, a variable pressure scanning electron microscope JEOL model 5900-LV (JEOL, Peabody, MA, USA) was used in order to avoid the sample coating by means of sputtering. The operational conditions used were a pressure of 20 Pa and an acceleration voltage of 15 kV; under these conditions, the charge of the samples was avoided because they are not conductive. The latest is based on the principle that the gas molecules in the chamber collide with the incident beam and with the electrons that leave the sample, these collisions become cations which are attracted by the negative charge of the sample and in that way the surface charge is neutralized. This is a good method for the analysis of non-conductive samples and avoids coating, especially if it is required to do microanalysis. It should be noted that we did not perform a porosity study, only surface morphology observation.

The contact angle of the P3HB scaffolds was measured by the sessile drop technique at room temperature (25 °C), putting 1 μL of distilled water on the scaffold surface. Images were acquired using a digital microscope with a camera (MicroView^®^, New York, NY, USA) at 25× and processed with image analysis software to calculate the angle between the water droplet and the surface [29].

### 2.7. Culture Conditions of the HaCaT Cells

Human HaCaT keratinocyte cells (ATCC: CSC-C8977H) were cultivated in culture medium DMEM-F12 (Dulbecco’s Modified Eagle Medium, GIBCO Invitrogen) supplemented with 10% Bovine Fetal Serum (BFS) at pH 7.2 ± 0.03. Cells were cultured and maintained at 37 °C with a CO_2_ concentration of 5% in an Eppendorf cell culture incubator.

### 2.8. Biocompatibility Analysis

Sterile scaffolds were carefully placed into 1 mL culture wells with a cell concentration of 5 × 10^5^ cells/mL and incubated at 37 °C with a CO_2_ concentration of 5% for 5 days. Thereafter, bright field images of the scaffolds were taken using a microscope to verify that cell adhesion had occurred. Then, the scaffolds were washed twice with 1 mL of PBS and dyed with Hoesch, acridine orange, and propidium iodide to count the viable and non-viable cells to estimate cell proliferation by epifluorescence microscopy.

## 3. Results and Discussion

### 3.1. Characterization of the P3HB Obtained

There are several methods reported in the literature regarding cell rupture based on halogenated and non-halogenated solvents for the extraction of P3HB from biomass. However, these studies showed that the use of compounds such as sodium hypochlorite and chloroform significantly affect the molecular characteristics of P3HB [17]. For this reason, in the first stage of this present study, P3HB obtained by extraction with ethanol–acetone was characterized.

Figure 1 shows the results of the physical appearance of the P3HB recovered by the acetone–ethanol method compared with the product obtained by the conventional method. The P3HB obtained by the hypochlorite–chloroform method (Figure 1A) was plastic and rigid; whereas the P3HB recovered using the method based on ethanol–acetone was odourless and had a soft texture, with particle sizes of from 0.074 mm to 5 mm (Figure 1B).

The recovery percentage using the method of hypochlorite–chloroform was 30%, while, the extraction method with ethanol–acetone was 79% (Table 1). Regarding the purity of the materials, the method of rupture and extraction using hypochlorite–chloroform was 99% and with the method of rupture and extraction with ethanol–acetone was 93% (Table 1).

Molecular weight represents an important parameter that determines the characteristics of biopolymers used for specific applications. In general, the P3HB should have a molecular weight >1 × 10^3^ kDa to satisfy the requirements for biomedical applications. In the present study, the molecular weight was affected by the extraction method used. For example, with the conventional method (hypochlorite–chloroform), a very low molecular weight was obtained (416 ± 68 kDa) (Figure 2, Table 1). However, using the ethanol–acetone method, P3HB had a high molecular weight of 1750 ± 98 kDa (Table 1). The values of molecular weight agree with those reported by Millán [27], who indicated that the hypochlorite–chloroform method causes the breakdown of the P3HB chains, affecting its molecular weight negatively, reducing it by up to 50%. This is due to the fact that the existence of native amorphous P3HB granules are quite vulnerable to alkaline saponification and rapidly decompose into soluble products, such as monomers and oligomers [30,31,32].

P3HB is more soluble in chloroform than other organic solvents. Therefore, after recovery of the polymer using the hypochlorite–chloroform method, a material with high purity (99%) was obtained. This value is consistent with those reported in the literature, which ranges from 98 to 100%. In comparison, the material recovered using ethanol–acetone had a purity of 93%. As is shown below, this lower purity was due to the presence of cellular and solvent residues as evidenced by the FTIR analysis.

Analysing the FTIR spectra of P3HB recovered by the different extraction methods, the characteristic peaks of the chemical structure of poly-3-hydroxybutyrate were found. For the product obtained by rupture and extraction with hypochlorite–chloroform, the spectra showed the same characteristic peaks of P3HB as previously reported, and had the same purity of 99% (Table 1). In contrast, the P3HB sample obtained by the ethanol–acetone method requires drying before characterization, to evaporate solvent residues. The spectrum of this sample showed bacterial residues, which agrees with what was obtained in the analysis of the purity. Table 2 shows the peaks CH_3_ (2974 cm^−1^), CH (2930 cm^−1^), C = O (1720 cm^−1^), and CH_2_ (1277 cm^−1^) characteristic of the P3HB polymer, which were observed in the materials recovered by the two extraction methods. However, for the P3HB obtained from the rupture and extraction method with ethanol–acetone, the OH groups (3283 cm^−1^), amides I (1647 cm^−1^), and amides II (1540 cm^−1^) were also present, in contrast to the polymer obtained from the extraction with chloroform.

The Young’s Modulus of the PH3B samples increased as a function of the molecular weight of the polymer (Table 1). Previous studies [12] indicated that Young’s Modulus of bacterial P3HB increased when the molecular weight of the polymer increased in the range of 240 to 1660 kDa. In polymers with a high molecular weight (more than 1660 kDa), there was a decrease in Young’s Modulus, because the polymers have significantly more amorphous regions due to their smaller degree of crystallinity, and therefore the modulus exhibits a significant reduction [12].

### 3.2. Morphology of the Scaffolds

The scaffolds generated by 3D printing from P3HB obtained by ethanol–acetone rupture and extraction showed surface roughness with protrusions (Figure 3) and presented some pores. The above characteristics would be considered a desirable property for the potential application of these scaffolds for biomedical applications as they could facilitate cell adhesion, propagation, and proliferation [29].

### 3.3. Determination of Contact Angle

The hydrophobicity of a surface can be determined by measuring the contact angle through the dispersion of a drop of water on the surface. The measurement of the contact angle on scaffolds generated in the present work by 3D printing using the P3HB obtained by the ethanol–acetone method yielded contact angles of 70° (Table 3), indicating that they were hydrophilic surfaces. In contrast, the contact angle of the scaffolds generated by electrospinning using the P3HB recovered by the rupture and extraction method with hypochlorite–chloroform (control) was 100 °. It is known that the smaller the contact angle (<90°), the more hydrophilic the surface, whereas, contact angles above 90° indicate a hydrophobic surface [12,28,33]. Studies have shown that cell adhesion, propagation, proliferation, and differentiation when surfaces are influenced by whether or not the surface is hydrophilic [34,35].

In the case of P3HB, the low contact angle is attributed to the fact that the hydroxyl (OH) and carbonyl (COH) groups present in this structure, depending on the spatial conformation of the material, tend to take an additional hydrogen ion (hydrogen acceptors), which generates a strong hydrophilic effect, coupled with the roughness of the surface, which favours lower contact angles [33].

### 3.4. Growth Kinetics of the HaCaT Cells

On the other hand, the characterization of the grown HaCaT cells was carried out using a DMEM-F12 medium supplemented with 10% SFB to determine the kinetic parameters of cell growth prior to their use with the scaffolds. Figure 4 shows that the exponential phase of cell growth in the DMEM-F12 medium with 10% SFB started after 72 h. It is important to point out that in all cases the cell viability was maintained at around 90% during the exponential growth phase.

The specific growth rate of the cell line in the DMEM-F12 supplemented with 10% SFB was 0.022 h^−1^, with a doubling time of 32 h, values that coincide with those reported in the literature for the cell line of HaCaT keratinocytes [36]. The resulting morphology under culture conditions was also consistent.

### 3.5. Biocompatibility Using HaCaT Cells

Figure 5 shows the epifluorescence micrographs for a scaffold designed with P3HB obtained from ethanol–acetone extraction by 3D printing (B) and its comparison with a scaffold obtained from electrospinning (A, control). On the FITC filter, viable HaCaT cells were observed that appeared in green when dyed with acridine orange; whereas, with the TRIC filter, no dead cells were observed (red colour). Lastly, the nuclei of living cells appeared in blue on the HOESCH filter, which indicates the absence of cytotoxicity of the biopolymer in the HaCaT culture. Figure 5A shows a heterogeneous growth of HaCaT cells on the control scaffold. This could be due to the contact angle being higher than 90° (100°) and the irregular shape obtained by electro-spinning. In contrast, a homogenous cell carpet is observed using the scaffold by 3D printing without pores (Figure 5B). In this case, the cell distribution was the best since the contact angle was less than 90°.

A more detailed observation, based on nuclei staining under the HOESCH filter (nuclei of living cells fluoresce blue), indicated that cell proliferation was consistent in all cases, but in the case of the 3D-printed scaffold, the cell distribution depends on the size of the pores, and a more homogeneous cell carpet is generated. It is important to note that the pore size of the scaffold is a determinant to induce cell proliferation, and, depending on the application, such a parameter should be evaluated. The above correlates with scaffolds generated using various techniques that provide supports with smooth surface morphologies, with certain protrusions, relief and some micropores, in which some cell lines have been cultivated. In the case of the culture of HaCaT cells, osteoblasts, neuronal cells, and fibroblasts, it was found that, with P3HB scaffolds, the cell proliferation was higher than with other polymers or combinations of polymers [28,37].

It is important to point out that, despite the purity of the P3HB obtained using the ethanol-acetone method being only 93%, it was found that this material did not cause any type of toxicity to the cells, allowing the HaCaT cells to adhere to all three of the scaffolds evaluated. Bacterial residues, do not imply the presence of live bacteria or complete microorganisms. Therefore, we used biocompatibility studies to evaluate if such residues were toxic and the biocompatibility studies demonstrated that there was no toxic effect. In addition, heat and chemical sterilizations were carried out to ensure that the cell culture could be developed under sterile conditions.

## 4. Conclusions

The novel ethanol–acetone extraction method used in this study produced P3HB with properties suitable for use as a scaffold in tissue engineering due to its high molecular weight. This kind of P3HB can be a material with better elastic properties for scaffold formation than low molecular weight P3HB [11,13]. The methodology used here is simple and the polymer obtained is of a quality to allow the growth of epidermal cells. In addition, the scaffolds generated by 3D printing had contact angles of less than 90°, showing a more hydrophilic surface compared to the control scaffold generated by electrospinning using P3HB obtained by the hypochlorite–chloroform method, which makes it an ideal material for its use in HaCaT cell growth.

The results reveal that on the scaffolds made with P3HB recovered by the extraction method with ethanol–acetone and manufactured by the 3D printing technique without pores, a higher cell proliferation was obtained when compared to the control scaffold.

This work shows that P3HB obtained by a simple ethanol–acetone extraction method works well as a biomaterial for the generation of scaffolds for application in epidermal tissue engineering. It also shows the influence of manufacturing parameters, such as the type of extraction or the method used for scaffold formation, on the properties of the obtained P3HB, and how they affect the proliferation of HaCaT cells, which are important model cells of human keratinocytes widely used in scientific research of skin tissue regeneration.

In summary, our results show the feasibility of using P3HB supports for the growth of epidermal cells and, thus, as a biomatrix for tissue engineering. At the same time, it is recognised that more work should be done on the optimization of the scaffolding design and its mechanical and functional characterization.

## Figures and Tables

**Figure 1 polymers-14-04021-f001:**
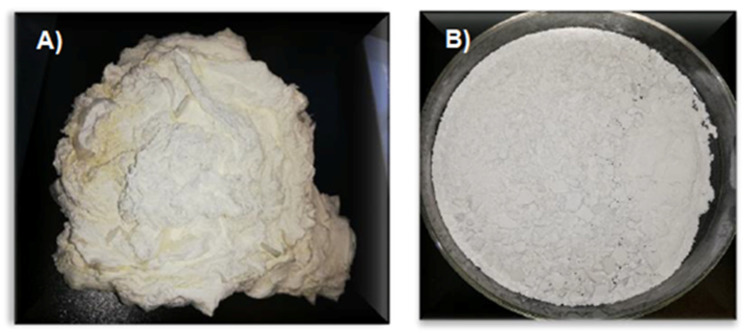
P3HB obtained using different extraction methods: (**A**) hypochlorite–chloroform, (**B**) ethanol–acetone.

**Figure 2 polymers-14-04021-f002:**
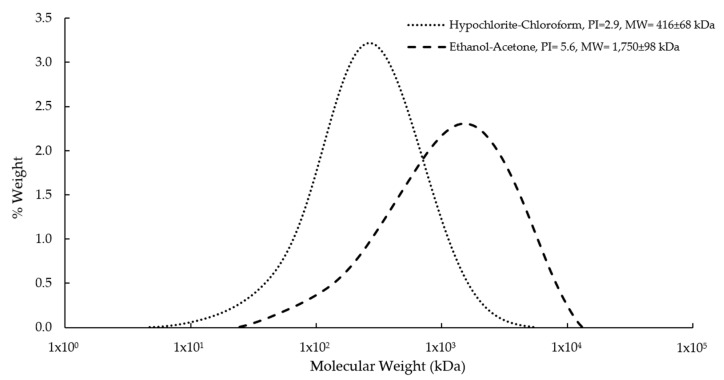
Molecular weight distributions of P3HB isolated from *A. vinelandii* OPNA cultures using ethanol-acetone and the conventional hypochlorite–chloroform method (Control).

**Figure 3 polymers-14-04021-f003:**
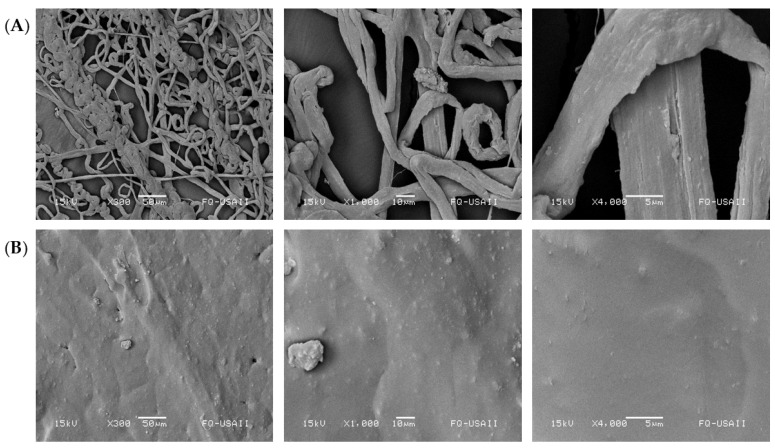
Morphological of the generated scaffolds, (**A**) the conventional method (Control), (**B**) the ethanol–acetone.

**Figure 4 polymers-14-04021-f004:**
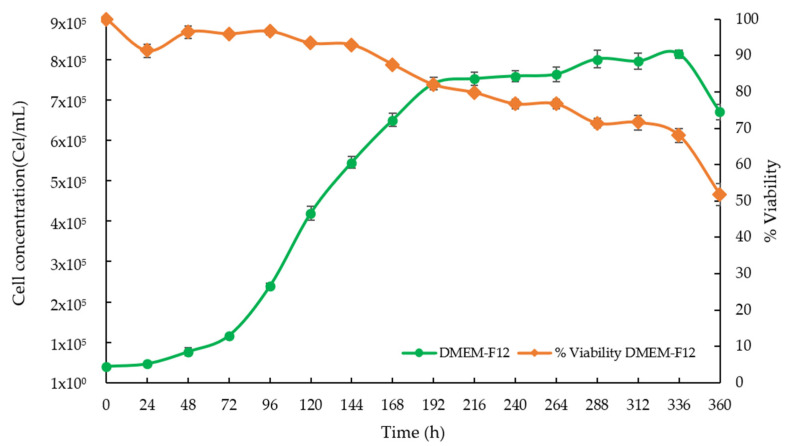
Growth kinetics of the HaCaT cell line in DMEM-F12 medium without scaffolds.

**Figure 5 polymers-14-04021-f005:**
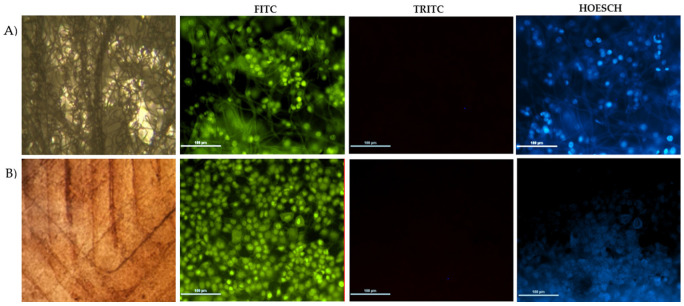
Biocompatibility of HaCaT Cells in scaffolds. (**A**) Control (electrospinning), and (**B**) 3D printing without pores.

**Table 1 polymers-14-04021-t001:** P3HB characterization.

Extraction Method	P3HB (kDa)	Purity (%)	Yield (%)	PI	Young’s Modulus (MPa)
Rupture and extraction with hypochlorite–chloroform	416 ± 68	99 ± 0.3	30	2.9	217
Rupture and extraction with ethanol–acetone	1750 ± 98	93 ± 1.0	79	5.6	261

**Table 2 polymers-14-04021-t002:** Identification of peaks present in the P3HB samples recovered by the two rupture and extraction methods.

Description	Ethanol-Acetone (cm^−1^)	Hypochlorite-Chloroform (cm^−1^)
OH group belonging to residual solvents	3282.8	N.A.
Crystalline CH3 asymmetric stretching	2973.1	2974.2
CH stretch	2931.2	2930.7
Carbonyl esters stretch (C=O)	1720.4	1719.8
Amide group I with C=O of protein-associated amides, which may contain C = C contributions from stretches of olefinic and aromatic compounds (residues from bacteria)	1647.6	--
Amide group II with NOH associated with proteins and may contain C=N contributions (bacteria residues)	1540.6	--
CH_2_ crystalline wagging (denoted helical structure)	1278.5	1275.1

**Table 3 polymers-14-04021-t003:** The contact angle of the scaffolds.

Extraction Method	Technique	Sample
Rupture and extraction with hypochlorite–chloroform (Control)	Electrospinning	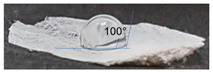
Rupture and extraction with ethanol–acetone	3D printing	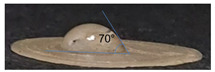

## Data Availability

The data presented in this study are available on request from the corresponding author.

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
