# Peer review of "Evaluation of Poly-3-Hydroxybutyrate (P3HB) Scaffolds Used for Epidermal Cells Growth as Potential Biomatrix"

_polymers, 2022, doi:10.3390/polym14194021_

Round 1
Reviewer 1 Report
The authors have done a commendable job by providing a well written manuscript with clearly defined objectives and detailed descriptions of the experiments conducted. I do have the following comments/suggestions:
1. Include images of the 3D printed and electrospun scaffolds with scale bars.
2. My understanding is that pores are formed on scaffolds when they are subjected to freeze drying. Also, it is imperative that there be an interconnected porous network with porosity not exceeding 70-80% so as to support cell proliferation and not compromise with the mechanical properties or the structural integrity of the scaffold. How were the 3D printed samples used for SEM without lyophilozing them and sputter coating them for them to be appear non porous? I think this experiment should be repeated for both samples and then the pore network should be compared between the two. Use ImageJ to determine the percentage porosity as has been done in literature.
3. If there were bacterial resides in the 3D printed scaffolds, does it not imply they are already contaminated? Were these scaffolds then used to culture cells?
4. I take it that the initial cell density was 5*10^5 cells/ml. Could you repeat this experiment by seeding the two types of scaffolds with the same cell density and using other techniques like MTT/XTT assays or flow cytometric analysis? I think it highly unlikely that there were no dead cells after 5 days of culture. It might be worth it to have additional assays like the ones I suggested to determine cell viability after 5 days of culture.
5. How many samples were tested? Were the experiments done in triplicate (n=3 samples for each study for both types of scaffolds)? Were the studies statistically significant?
6. The article needs to be proofed using Grammarly or a similar tool to get rid of syntaxes and grammatical errors. Line 110 is incomplete.
Reviewer 2 Report
Dear Authors
Your article is very attractive, novel, and well-organized. Obtaining scaffolds and tissue engineering is a topic interesting for different types of professionals and I believe that will attract a wide range of readers.
However, there are some points in the manuscript that could be improved.
English is appropriate and except some minors check is satisfying for the article.
The introduction part should be improved especially to involve novel and up-to-date references. As it is now, most of the references are older than 5 (or 10 years) ago.
The experimental section is well described except for some missing points (please see the attached version of the article with suggestions that could improve it).
The results section is suitable and consequent after the Experimental section, and the Conclusion part could be extended and at the end should involve some future/vision about this area, or about the material - e.g. how to improve it and adjust to the proper application(s).
Thank you.

Round 2
Reviewer 1 Report
Thanks for addressing all my comments. Good job!